# Fabrication and Characterization of Hydrogen Peroxide and Thymol-Loaded PVA/PVP Hydrogel Coatings as a Novel Anti-Mold Surface for Hay Protection

**DOI:** 10.3390/polym14245518

**Published:** 2022-12-16

**Authors:** Eyal Malka, Ayelet Caspi, Reut Cohen, Shlomo Margel

**Affiliations:** 1Chemistry Department, Bar-Ilan University, Ramat Gan 5290002, Israel; 2TAMA Group, Kibbutz Mishmar HaEmek, Tel Aviv 1923600, Israel

**Keywords:** anti-mold, hydrogel, biodegradable, coatings, preservatives, thymol

## Abstract

Animal food source production is increasing due to the growing world population. Many sources (e.g., hay) are prone to mold development, resulting in food degradation. This study proposes an environmentally friendly anti-mold fungicide comprising hydrogen peroxide (HP) and thymol entrapped in a polyvinyl alcohol/pyrrolidone (PVA/PVP) hydrogel (PVA is biodegradable and PVP is water soluble and non-toxic) coated on a polyethylene (PE) films for preservative hay packaging. The hydrogels improved the thermal stability of the entrapped HP and thymol, resulting in a prolonged release into the hay and thereby increasing anti-mold activity. The hydrogel composition and morphology, thymol and HP thermal stability, and release rates through indirect (gas phase) contact were investigated. Fungicidal capabilities were tested, indicating wide-range efficiency against mold growth on hay with a clear advantage for the thymol-loaded hydrogels. No visual side effects were observed on hay exposed to the released fumes of HP and/or thymol. These results demonstrate the potential of thymol-loaded hydrogels as effective and safe post-harvest preservatives.

## 1. Introduction

The consistent growth of the world population is accompanied by an increase in food intake, including meat and milk products from animal sources [1]. These livestock depend heavily on harvest products as part of their daily nutrition. Farmers pack the products (hay and silage) into bales in the field and leave them outside for several months. During this time, they are exposed to weathering and are prone to mold development. Such losses and decreases in crop quality lead to animal feed shortage [2]. Moreover, mold spores and mycotoxins released from infected hay bales can cause severe illness and death to the farmers and livestock [3,4].

To prevent mold development on hay, various methods are used, such as dehydration, inoculation by competing microbiomes and sterilization [2]. However, these hay preservation techniques have partial efficiencies, are limited in terms of cost-benefit considerations, or contain hazardous chemicals. Hay dehydration requires machinery and a climate-controlled storage facility, which makes it expensive [5]. Microbiome inoculation involves a hay-friendly bacterium, which thrives favorably and dominates over the mold growth, but, in practice, the results are unsatisfactory with only partial efficiency [6]. Sterilization additives such as sprayed ammonia are considered effective fungicides but involve corrosion and inhalation hazards [6]. These challenges led to the search for more economical, effective, safe materials and techniques for animal feed preservation.

A promising alternative preservative is essential oils, well-known natural fungicides with low toxicity, which are safer than synthetic chemicals and used in food products [7]. Our study focuses on thymol, a volatile essential oil that has been widely investigated and used as an additive in the agrochemical and food industries [8]. The antifungal mechanism of thymol is based on hydrophobic interactions with the cell membrane, resulting in its destruction [9]. Hydrogen peroxide (HP) was chosen as an additional fungicide due to its proven anti-mold properties [10], low toxicity, safety and lack of permanent residual material [11]. The mechanism of action of HP is based on hydroxyl radical formation by the Fenton reaction, where free radicals lead to general oxidative cell damage [11].

Polyvinyl alcohol and pyrrolidone (PVA/PVP) hydrogel were chosen for entrapping and releasing HP and thymol for the following two reasons: (1) HP and thymol can form various hydrogen bonds with the hydrogel, entrapping them efficiently within the matrix (Figure 1); these interactions stabilize the entrapped materials and result in their controlled release. (2) The hydrogel has a high tensile strength, although its interactions are based solely on weaker physical crosslinks rather than stronger chemical interactions [12]. These non-covalent and reversible physical interactions allow the PVA to biodegrade over time [13] and the non-toxic, water-soluble PVP to dissolve into the soil [14]. Therefore, the PVA/PVP hydrogel is a non-toxic, environmentally friendly material.

In our work, the hydrogel coating is applied to the PE surface [15] to further test it against mold growth. The interaction between the hydrogel coatings and the hay occurs via HP and thymol vapor contact. Previous studies investigated HP and thymol pesticide activity by vapor and direct contact and found a clear advantage to the former [11,16].

Our study proposes HP and thymol-loaded PVA/PVP hydrogel coatings on PE as a novel anti-mold surfaces for hay protection. The advantages of this anti-mold technology are as follows: (1) the PVA/PVP hydrogel can be applied to polymeric films as a coating, providing a homogeneous and orderly anti-mold exposure to the hay. (2) The hydrogel stabilizes the fungicidal material, allowing its controlled release and prolonged duration in the hay, resulting in higher efficiency. (3) PVA and PVP are biodegradable and non-toxic, respectively, making this hydrogel environmentally friendly. (4) HP is an environmentally friendly pesticide that decomposes into oxygen and water after use, while thymol is a natural and non-toxic pesticide.

Previous work highlights the benefits of food preservatives based on essential oils entrapped in polymers with encouraging anti-microbial results. Thymol-loaded nanofiber mats [17] show high efficiency against mold. The Mamdouh group encapsulated peppermint and green tea essential oils in chitosan nanoparticles [18] with promising results against gram-positive and gram-negative bacteria. The Yoksan group synthesized carvacrol-loaded chitosan nanoparticles [19], shows antibacterial properties. These studies enhance the feasibility of our approach.

It was of interest to investigate the contribution of the PVA/PVP hydrogels to HP and thymol thermostability as well as their morphological effects on the hydrogels. The release profiles of HP and thymol from the hydrogels into the hay environment were also investigated, as well as their anti-mold activity. Our main tool for characterizing the hydrogel molecular properties was ATR-FTIR. HP content and release rate were determined by potassium permanganate titration and Quantofix^TM^ peroxide test strips, while thymol content and release rate were determined by UV-Vis spectrophotometry. SEM and AFM were used to evaluate hydrogel microstructure morphology and average roughness, respectively. Free and entrapped thymol thermostabilities were compared by TGA-MS analysis, while those of HP were compared by an isothermal test. Anti-fungicidal tests were conducted in a humidity- and temperature-controlled incubator. This approach has wide applicability.

## 2. Materials and Methods

### 2.1. Materials and Reagents

Polyvinyl alcohol (PVA), average MW 89–98 kDa, 99% hydrolyzed (Sigma-Aldrich., Saint Louis, MO, USA), polyvinylpyrrolidone (PVP K30), average MW 40–80 kDa (ISP., Wayne, NJ, USA), urea-hydrogen peroxide (UHP), 97% (Alfa Aesar., Heysham, Lancashire, UK), thymol >98.5% (Sigma-Aldrich., Saint Louis, MO, USA), ethanol (99%), AR (Bio-Lab., Jerusalem, IL, USA), and sulfuric acid (96% for analysis-ISO) (Carlo Erba., Emmendingen, Germany). Potassium permanganate, analytical grade reagent, 99.5% (Fisher chemical., Pittsburgh, PA, USA), sodium oxalate powder (ACS, 99.5+%) (Alfa Aesar., Heysham, Lancashire, UK), and Quantofix™ Peroxide 25 and 100 semi-quantitative test strips (Mancherey-Nagel., Düren, Germany).

### 2.2. Fabrication of PVA/PVP Hydrogels

PVA/PVP hydrogel was prepared by heating PVA aqueous solution at 95 °C until dissolution followed by addition of PVP and cooling to room temperature. Briefly, 3.75 g of PVA was dissolved in 19.75 g of double-distilled water (DDW) at 95 °C; 1.5 g of PVP was added, and the solution was stirred for 1.5 h and cooled to 78 °C (until the foam from the reaction disappeared). PVA/PVP/HP hydrogel solution was prepared by addition of UHP into the PVA/PVP hot (78 °C) aqueous solution and brief mixing until complete dissolution (Figure 1A). PVA/PVP/thymol and PVA/PVP/HP-thymol hydrogel solutions were prepared by addition of thymol and HP combined with thymol, respectively, into the PVA/PVP aqueous solution until a homogeneous white emulsion was obtained (Figure 1B,C). HP and thymol contents were varied according to the desired concentration.

### 2.3. Hydrogel Gelation and Coating Process

Hydrogel coatings were prepared on PE films. The PE was treated with corona (VETAPHONE Corona & Plasma, Kolding, Denmark) (350 W·min/m^2^, 20 scans) for surface oxidation [20,21]. Immediately after mixing, the hot hydrogel aqueous solution (78 °C) was poured into the corona-treated PE surface, followed by “Mayer rod” coating [22] with a rod blade size of 400 microns (wet thickness). The hydrogel-coated PE surface was gelated by the freeze-thaw method [23]. The coated PE films were refrigerated overnight at −18 °C and thawed (one freezing-thawing cycle).

### 2.4. Characterization

#### 2.4.1. Hydrogel Coating Molecular Characterization

Fourier transform infrared/attenuated total reflectance (FTIR/ATR) spectra of PVA/PVP hydrogels were obtained at room temperature using an Alpha-FTIR Quicksnap^TM^ sampling module equipped with a Platinum ATR diamond module (Bruker, Germany) and translated by the OPUS program. The absorbance measurements were set to the range of 500–4000 cm^−1^.

#### 2.4.2. Determination of Thymol Content in Hydrogel Coatings

##### PVA/PVP/Thymol Hydrogel Coating

The hydrogel coating (10 mg) was peeled from the PE-coated surface, inserted into a centrifuge tube prefilled with 20 mL of analytical grade ethanol and centrifuged (6500 rpm, 30 min); an additional centrifugation with fresh ethanol was applied to verify complete thymol extraction. Thymol absorption (λ = 276 nm max) was monitored with a 1E UV/visible spectrophotometer (Varian Cary., Cary, NC, USA). Extracted concentrations were calculated using a pre-calibrated curve of thymol (dissolved in ethanol). Thymol content was measured in triplicate.

##### PVA/PVP/HP-Thymol Hydrogel Coating

The hydrogel coating (10 mg) was peeled from the PE-coated surface and centrifuged (6500 rpm, 30 min) in 20 mL of DDW in a prefilled tube. The extracted HP aqueous solution contains also partially extracted thymol, which was quantified with a calibration curve (*λ* = 273 nm max absorption) by a Varian Cary 1E UV/visible spectrophotometer. For complete thymol extraction, the hydrogel coatings were centrifuged (6500 rpm, 30 min) in a tube with 20 mL of analytical-grade ethanol until complete thymol extraction (verified by consecutive extraction with fresh ethanol). The thymol concentration was determined by a calibration curve of thymol dissolved in ethanol. The total thymol concentration includes both aqueous and ethanol-extracted solutions.

#### 2.4.3. Determination of HP Content in Hydrogel Coatings

##### PVA/PVP/HP Hydrogel Coating

The hydrogel coating (10 mg) was peeled from the PE-coated surface, dipped in a vial containing 10 mL of DDW and shaken at room temperature until all of the HP was extracted into the water (a quick vigorous manual shaking for a few seconds). To ensure that no HP remained entrapped, a sequential extraction with fresh water extracted the remaining material. The concentration of released HP was determined (in triplicate) by adding 550 µL of H_2_SO_4_ (96%) to the vial and titrating it with a pre-calibrated KMnO_4_ solution [24].

##### PVA/PVP/HP-Thymol Hydrogel Coating

The potassium permanganate titration method is not applicable to hydrogels that combine thymol and HP as both are oxidized by KMnO_4_. Therefore, the extracted HP was quantified by test sticks specific to peroxides. The hydrogel coating (10 mg) was peeled from the PE-coated surface. The peeled coating was centrifuged (6500 rpm, 30 min) in 20 mL of DDW in a prefilled tube until complete HP extraction. The test sticks are semi-quantitative visual colorimetric indicators with a wide peroxide measuring range (1–100 mg/l) and are pre-calibrated and accurate at a pH range of 2–9 and a temperature range of 4–30 °C. The observed color is compared to the color scale bar. HP content was measured in triplicate.

#### 2.4.4. Release Rate System

To determine HP and thymol vapor release rate, Petri dishes were filled with 3 g of hay and sprayed with water (4 g). For preventing direct contact between the hydrogel coating and the hay, the dishes were wrapped with a plastic net separator (Figure 2A). The hydrogel-coated PE films (Figure 2B) were cut to the diameter and circle shape of the Petri dish (100 mm). The pre-cut coated films were placed on the net with the hydrogel coating oriented to the hay direction (Figure 2C), while the PE film side is oriented in the opposite direction in such a way that it is in contact with the Petri dish cover. The dishes were incubated at 27 °C with 60% humidity.

##### Thymol Release Rate

PVA/PVP/HP-thymol hydrogel coatings (10 mg) were peeled from the PE-coated surface. The peeled coatings were centrifuged (6500 rpm, 30 min) in a 20 mL DDW-filled centrifuge tube. The absorption of the extracted thymol aqueous solutions (*λ* = 273 nm max absorption) was determined as described above. The coatings were centrifuged (6500 rpm, 30 min) in a tube refilled with 20 mL analytical grade ethanol until complete thymol extraction (verified by consecutive extraction with fresh ethanol). For PVA/PVP/thymol, a similar procedure was applied; thymol concentrations were measured by the calibration curve of thymol dissolved in ethanol. PVA/PVP/HP-thymol total extracted thymol was obtained by combining the aqueous and ethanol absorbance, while PVA/PVP/thymol total extracted thymol derived solely from the ethanol solution. Subtraction of remaining thymol concentrations at predetermined times (after 2, 5, 12 and 26 days) from the initial value provided the released thymol (division by the initial concentration gave the relatively released thymol).

##### HP Release Rate

PVA/PVP/HP-thymol hydrogel coating (10 mg) was peeled from the PE-coated surface and centrifuged (6500 rpm, 30 min) in a tube with 20 mL distilled water until complete HP extraction. HP concentrations in the aqueous solution were determined by inserting peroxide test sticks into the HP extraction solutions and comparing the obtained color to the scale bar. Each sample provides the remaining HP concentration, which was subtracted from the initial concentration to give the released HP concentration. The released concentration divided by the initial concentration yielded the relative released concentration. HP was monitored in triplicate at the beginning and after 2, 5, 12 and 26 days. PVA/PVP/HP hydrogel coating (10 mg) was peeled from the PE-coated surface, dipped in vials containing 10 mL of DDW and shaken at room temperature until the HP was completely extracted as described in Section 2.4.3. The released HP concentration was determined by adding 550 µL of H_2_SO_4_ (96%) and titrating with a pre-calibrated KMnO_4_ solution. The released concentrations were obtained as described above for PVA/PVP/HP-thymol (in triplicate).

#### 2.4.5. Morphological Characterization

The effect of the entrapped thymol and/or HP on the surface morphology of the hydrogel coatings was revealed by environmental scanning electron microscopy (E-SEM) (JSM-840., Tokyo, JP with magnification of 200, 2000 and 10,000. Morphological characterization was conducted under low vacuum pressure (9.5 mm) and under voltage of 5 kV. Sizes of morphological formations associated with the entrapped material/s were calculated by the image analysis program Image J.

#### 2.4.6. Surface Roughness

The effect of entrapped HP and/or thymol on the hydrogel morphology, in particular the surface roughness, was investigated by AFM. The analysis was carried out using a Bio FastScan scanning probe microscope (Bruker AXS., Billerica, MA, USA). Images were obtained using soft tapping mode with a Fast Scan B (Bruker) silicon probe (spring constant of 1.8 N/m). The cantilever’s resonance frequency was approximately 450 kHz (in air). Measurements were performed under environmental conditions. Images were captured in the retrace direction with a scan rate of 1.6 Hz and resolution of 512 samples/line. The “Nanoscope” analysis software was used for image processing and roughness analysis (before analysis, the “flatten” and “planefit” functions were applied). The image scans were performed on a 5 µm scale. The roughness average (Ra) was used as the main parameter to distinguish between samples. The hydrogels were freeze-dried (lyophilized).

#### 2.4.7. Thymol Thermal Stability in PVA/PVP Thymol and HP-Thymol Hydrogel Coatings

Thermogravimetric analysis gas chromatograph mass spectrometry (TGA-GC/MS) experiments were performed with a (PerkinElmer., Waltham, Middlesex, MA, USA) ‘Pyris 1’ TGA, ‘Clarus 680’ GC and ‘Clarus SQ 8 C’ MS instruments. Samples were subjected to a TGA oven with a heating rate of 20 °C/min from 30 to 630 °C under nitrogen atmosphere (balance purge 80 mL/min: sample purge 20 mL/min) on alumina crucibles. The TGA was connected to a transfer line (TL9000) of ‘red shift’ heated to 280 °C. All gases were emitted from the TGA oven flow through a heated capillary without any separation into the mass spectrometer. The conditions were as follows: electronic impact ionization at 70 eV, source temperature of 220 °C, GC transfer line temperature of 250 °C and mass range of 30−300 Da with 0.2 s dwell time. The data were collected and analyzed by PerkinElemer GC/MS TurboMass software.

#### 2.4.8. HP Thermal Stability in PVA/PVP/HP Hydrogel Coating and Water

To evaluate the hydrogel contribution to HP thermal stability, HP was compared with HP entrapped in PVA/PVP hydrogel, both containing 10 w% HP. As the TGA-MS detector cannot distinguish between the oxygen that originates from HP decomposition products and water vapor, the potassium permanganate titration method [25] was applied. PVA/PVP/HP hydrogel coatings with size and weight of 4 cm^2^ and 59 ± 4 mg, respectively, and HP solutions with weight of 66 mg were incubated in an air-forced laboratory oven preset to isothermal 82 °C for accelerated HP decomposition [26]. The remaining HP was subtracted from the initial value to give the decomposed HP. Results were divided by the initial HP content to give the relative decomposed HP at the beginning and after 0.5, 1, 2, 3, 4, 5, 6, 9, 16.5 and 24 h.

#### 2.4.9. Determination of Anti-Mold Properties

The effectiveness of PVA/PVP HP, thymol and HP-thymol hydrogel coatings against mold growth was evaluated by allowing optimal mold growth conditions in Petri dishes containing 3 g of hay and sprayed water, followed by stretching and wrapping nets on the Petri dish top to allow the HP and thymol vapor to spread towards the hay environment. The sides of the hydrogel-coated PE sheets were placed on the stretched nets and covered with the Petri dish top. The plates were incubated for 26 days at 27 °C and 55–60% humidity. Mold growth was monitored and evaluated by measuring the mold size (cm^2^) over time. Mold fully spread on hay is considered 100% growth (PVA/PVP hydrogel coating served as a control).

## 3. Results and Discussion

### 3.1. FTIR/ATR Spectra of PVA/PVP Hydrogels

FTIR/ATR spectroscopy was used to assess the chemical composition of the hydrogels. The characteristics of PVA/PVP and PVA/PVP/HP (Figure 3 blue and red, respectively) were investigated by FTIR/ATR and compared in our recent manuscript [25]. The addition of thymol to the hydrogel significantly contributes to hydrogen bond interactions, as the thymol phenolic hydroxyl group [27] (O-H stretching at 3300–3360 cm^−1^) can interact with the PVP carbonyl or the PVA hydroxyl group (Figure 1B, C). Thymol molecular structure analysis by FTIR is well known [27]; however, when thymol is entrapped in the hydrogel, its vibrations shift to a higher wavenumber (Figure 3 black). The entrapped thymol vibrations at 846, 1091, 1427, 1653, 2919 and 2944 cm^−1^ are assigned to C-H wagging out of the plane, 1:3:4-substitution, isopropyl symmetric and asymmetric bending, C=C stretching and C-H symmetric and asymmetric stretching, respectively [27]. The vibration shifts (40 shifts) of the entrapped thymol are explained by its possible interactions with PVA and PVP. A similar pattern of results was obtained in previous studies on thymol in emulsions or coated with polymers [28,29]. The observed thymol vibrations confirm its presence in the hydrogel matrix. PVA/PVP/HP-thymol (Figure 3 orange) has a shoulder (2830 cm^−1^) attributed to OH stretching unique to HP [30], confirming its presence. The thymol peaks at 3000–3500, 2850–3000 and 847 cm^−1^ are assigned to OH stretching, –CH symmetric stretching and asymmetric stretching of –CH out of the plane, respectively. These peaks decrease dramatically, while the peak assigned to the C-O bond at 1026 cm^−1^ grows and shifts (−65 shifts). These vibrations [27,31] and shift differentiation support a possible known reaction between HP and thymol to form thymoquinone [32]. Moreover, the weaker C-O bond and stronger –OH stretching signals can indicate the transformation of the thymol phenol into thymoquinone ketone, which is accompanied by a corresponding decrease in the thymoquinone –CH signals (Figure 3).

### 3.2. Entrapped HP and/or Thymol Content

The measured HP content in the PVA/PVP/HP hydrogel (5 w% HP precursor) was 15.5 ± 0.6 w%. The measured thymol content in thymol-loaded hydrogels (5 and 1.25 w% thymol precursors) was 17.8 ± 2.4 and 4.0 ± 0.5 w%, respectively. The HP and thymol contents of the combined coating are shown in Table 1.

The actual HP and thymol concentrations in the coatings are greater than those in the precursor solutions, indicating hydrogel water loss. The results support preferable interactions of HP with the PVA/PVP polymers rather than with water. These preferable HP interactions can be explained by its more branched hydrogen bonds [26] with the polymers. The higher thymol content in the hydrogel is possibly due to its size. A reasonable explanation is that the much larger thymol molecules (thymol and water molecular weights of 150.22 and 18 g/mol, respectively) are caged with less ability to release from the branched hydrogel matrix.

### 3.3. HP and Thymol Cumulative Release Rates from Loaded PVA/PVP Hydrogel Coatings

Release rate tests were conducted to define the HP and thymol release profiles. Moreover, we sought to understand the effects of different HP and thymol concentrations, solely and in combination. Thymol release rates were tested on PVA/PVP/thymol coatings (Figure 4 black) with an average weight of 10.0 ± 2.1 mg and thymol content (w%) of 17.8 ± 2.4. The hydrogel coatings demonstrated a fast release of thymol within the first five days (70% cumulative release), after which the thymol release curve moderately decreased until the 12th day, followed by a mild release rate up to the 26th day. The decrease in the release curve indicates that the coatings regain some of the thymol from the environment. The thymol regained in the coating may be due to the closed Petri dish, which provides a closed system. The thymol vapor releases until equilibration are achieved between the hydrogel coating and its close environment. Continuous release of thymol vapor may form oversaturation. Therefore, the system equilibrates the excess thymol vapor by absorbing it back into the coating.

HP release rates were tested on PVA/PVP/HP hydrogel coatings (Figure 4 red) with an average weight of 10.0 ± 0.1 mg and HP content (w%) of 15.5 ± 0.6. The coatings demonstrate a fast release within the first two days, after which the curve moderately increases until all HP is released after eight days. The faster release relative to thymol can be explained by the low stability of HP in the tested temperature (27 °C). Once HP releases from the coating, the unstable molecule decomposes into water and oxygen [33], and as a result, fewer HP molecules are available and equilibrium is less likely, leading to a continuous release until completion.

HP and thymol release rates were measured on PVA/PVP/HP-thymol hydrogel coatings (Figure 4 orange and yellow), with an average weight of 10.0 ± 0.4 mg. The HP and thymol contents (w%) were similar 5.6 ± 0.5 and 5.9 ± 0.5, respectively. The hydrogel coatings demonstrate fast release within the first two days, after which the curve moderately increases until all HP is released after three days. About 95% of the thymol is released within five days, followed by a mild thymol reload into the coatings (~10%) during the consecutive eight days, and a moderate release until the 26 day. HP and thymol release rates were increased when combined in the hydrogel (Figure 4 orange and yellow), possibly due to incompatible interactions between the hydrophilic HP and hydrophobic thymol that cause repulsion between the two molecules, which facilitates their release. In conclusion, coatings loaded solely with HP or thymol performed favorably due to the extended and moderate release. Coatings with both molecules had a less favorable faster release with a sharper release profile.

Mold growth was monitored simultaneously. The PVA/PVP/HP coatings (Figure 4 red) show low activity, while both thymol-loaded coatings (Figure 4 black and orange) achieve complete mold inhibition. This is in accord with the shorter duration of HP release (Figure 4 yellow), which decreases the exposure of the mold. These findings led us to abandon further research on the less effective PVA/PVP/HP and focus on the thymol-bearing coatings. These coatings were fabricated with decreased HP and thymol content—1.25 w% of precursor (2.2 ± 0.4 w% actual) for thymol alone and 0.63 w% each for HP and thymol in PVA/PVP/HP-thymol (0.66 ± 0.01 and 0.95 ± 0.04 w% actual, respectively). The lower content of thymol and HP was chosen to reveal the contribution of HP to the anti-mold activity.

HP and thymol release rates and mold growth tests were performed in parallel to find any correlation between the release rate and mold growth. Moreover, the effect of HP combined with thymol on the release rate was studied (Figure 5). HP and thymol (0.63 w% each) were completely released from PVA/PVP/HP-thymol within two days (Figure 5 yellow dotted and orange dashed lines, respectively), while about 90% of the thymol (1.25 w% of precursor) was released from PVA/PVP/thymol coatings within two days, followed by thymol regain (20%) until the tenth day, then reversed until all of the thymol was released on the 24 day (Figure 5 black). This release profile indicates the formation of a thymol-saturated environment and its equilibrium with the hydrogel coatings.

Hay exposed to PVA/PVP/HP-thymol coatings yielded partial mold inhibition (mold appeared on the sixth day), while PVA/PVP/thymol coatings achieved complete inhibition. These findings can be explained by the insufficient HP and thymol concentrations released from PVA/PVP/HP-thymol coatings into the hay environment. In contrast, PVA/PVP/thymol induces a saturated thymol environment. The excess thymol eradicates the mold spores, thereby preventing any growth. In conclusion, the release profiles support the anti-mold results. Moreover, thymol combined with HP anti-mold performance was found to be inferior. Thus, PVA/PVP/thymol hydrogel coatings are preferred; 1.25 w% of thymol was found to be the minimal effective concentration for mold eradication.

### 3.4. Effect of HP and/or Thymol on PVA/PVP Hydrogel Coating Morphology

PVA/PVP hydrogel neat coatings introduce a rough and porous surface (Figure 6A–D) with a 0.77 ± 0.17 µm average opening pore size diameter, while the HP-loaded ones have a smooth texture with a round shape with an average size of 1.27 ± 0.44 µm (Figure 6G). The smoother surface of the PVA/PVP/HP hydrogel (Figure 6E–G) is due to oxidation reactions between HP and PVA [13].

During hydrogel preparation, the heat decomposes HP into oxygen and water molecules [33]. The hot oxygen gas molecules penetrate through the hydrogel surface and form holes. Figure 6H verifies the presence of thymol particles entrapped in the PVA/PVP hydrogel coatings. Moreover, the coatings were covered homogeneously with thymol particles with an average size of 146 ± 57 µm. The entrapped thymol particles introduce a hollow web (Figure 6I) with an inner disc and flake shape (Figure 6J). The PVA/PVP/HP-thymol hydrogel has a smooth texture with a dense net fiber shape (Figure 6K) that differs from the thymol particle shape or the holes, caused by the HP. The inner view exposes a continuous rod shape (Figure 6L) with an average width of 1.43 ± 0.23 µm.

A possible explanation for the formation of the rod-shaped structures is the favorable interactions of the hydrogel with the hydrophilic HP rather than the hydrophobic thymol. This may cause the thymol to be expelled onto the hydrogel surface. The resulting concentrated thymol on the surface can agglomerate into rod-shaped structures. The addition of HP and/or thymol to PVA/PVP hydrogel contributes to the formation of different coating morphologies. The sizes of the thymol particles (Figure 6H–J), the rod-shaped product of HP and thymol (Figure 6K–M), the hydrogel pores (Figure 6D) and the HP holes (Figure 6G) were calculated by the public domain software Image J.

### 3.5. Effect of HP and/or Thymol on Hydrogel Coating Roughness

SEM images of the hydrogel coatings illustrate different morphologies. A complementary quantitative evaluation of the average roughness provides a deeper understanding. Each coating was scanned by AFM. PVA/PVP has a smooth (*R_a_* = 16.6 ± 2.4 nm) mildly hill-shaped texture (Figure 7a,b). PVA/PVP/HP has a flattened (*R_a_* = 4.5 ± 2.5 nm) perforated surface with round shapes seen in the SEM image (Figure 6G) identified as holes by AFM (Figure 7c,d). These holes could have been formed by oxygen molecule gas bubbles originating from the hydrogen peroxide [26], which decomposed in the heated hydrogel solution. PVA/PVP/thymol (*R_a_* = 39.1 ± 18.1 nm) has ridge shapes; the thymol microstructures anchored into the hydrogel surface may contribute to the higher roughness (Figure 7e,f). The addition of HP combined with thymol into the hydrogel formed new rod-shaped microstructures in the SEM image (Figure 6L,M) completely different from the hydrogels loaded solely with HP or thymol. These new micro-structures result in a lumpy morphology and steep inclines), with the highest roughness (*R_a_* = 121 ± 55 nm) among the four hydrogels(Figure 7g,h). This significantly higher roughness is possibly a result of the prominent thymol contribution, while HP perforation contributes to the increased roughness amplification. From these results, HP, and thymol both have a significant contribution to the hydrogel micro- and nanoscale morphology.

### 3.6. Hydrogel Entrapping Effect on Thymol Thermal Stability

The analysis was conducted to study the stabilizing effect of the PVA/PVP hydrogel under heat treatment on entrapped thymol relative to pure thymol grains. TGA-MS provides a “one-size-fits-all” analysis of what is released during heating in the TGA oven. Upon exposure of the sample to a thermal process, sometimes the compounds evaporate, while in other cases they decompose during pyrolysis. Here, the first case is relevant, and we detect thymol itself. While standard TGA usually provides information about the thermal stability of the sample, the hybridization with the mass spectrometer provides an indication of the temperature range in which the thymol is released. The thermal stability of entrapped thymol was evaluated relative to neat thymol grains. Thymol was identified by recording with an MS detector, which tracks its molecular weight (~150 g/mol) and base peak (~135 g/mol). Pure thymol loss occurs during one step, beginning after 2 min (70 °C) and ending after 5 min (130 °C). The continuous thymol reading after 5 min in the chromatogram is a result of thymol residues retained on the MS detector and is not due to slow release. This is confirmed by the thermogram, which attains zero weight percentage after 5 min (Figure 8A).

The thymol entrapped in PVA/PVP hydrogel lost weight in two steps. The first step begins after 14 min (310 °C) and ends after 20 min (430 °C), while the second step begins after 22 min (470 °C) and ends after 25 min (530 °C, Figure 8B). For the thymol entrapped in PVA/PVP/HP-thymol, weight loss occurs in one step from 21.5 (460 °C) to 25 min (530 °C, Figure 8C). The addition of urea hydrogen peroxide (UHP) seems to affect the initial release of thymol. If we assume that the initial release is due to thymol on the surface, the combination with UHP changes the surface (Figure 6K–M and Figure 7g,h), therefore the thymol trapped inside may be the only one that remains and is released in the second stage. Figure 8D summarizes each thermogram with a bidirectional arrow indicating the thymol weight loss region as a function of time. The results indicate a far more thermostable entrapped thymol relative to its pure form. Thymol entrapped in PVA/PVP or PVA/PVP/HP hydrogels showed a similar thermostability (Figure 8D red and orange, respectively). Therefore, HP did not have much effect. Previous studies, which demonstrate that polymeric cases enhance the thermal stability of essential oils [18,19], corroborate our findings.

### 3.7. PVA/PVP Hydrogel Entrapping Effect on HP Thermal Stability

We hypothesized enhanced thermal stability of HP entrapped in a hydrogel over HP aqueous solution due to branched hydrogen bond interactions between HP and the hydrogel (Figure 1A). To confirm our assumption, a stability test was conducted under isothermal conditions (82 °C), as HP decomposition accelerates at higher temperatures [34]. The high sensitivity of HP to decomposition makes it easier to distinguish between the thermal stability of HP entrapped in a hydrogel (Figure 9 blue) and free HP (Figure 9 orange). The results confirm our expectations. HP entrapped in the hydrogel shows a moderate rate of decomposition (Figure 9 blue), which lasts almost 17 h, while HP in solution completely decomposes within 5 h (Figure 9 orange). This significant thermostability advantage over an aqueous solution proves that the PVA/PVP hydrogel is a well-stabilizing HP carrier.

### 3.8. Anti-Mold Activity of HP and/or Thymol Loaded Hydrogel Coatings

The test was performed in Petri dishes. The coated PE sheets were located on the dish top, and hay was placed on the bottom. Spacer nets were placed beneath the coated PE sheets and the hay (Figure 2). Our questions dealt with the relative anti-mold activity of each hydrogel coating and the feasibility of HP and thymol synergism. The mold growth exposed to the various coated sheets was monitored. PE coated with PVA/PVP (control) shows a complete mold development within 26 days (Figure 10 blue). PVA/PVP/HP (5 w%) coatings introduce partial mold inhibition followed by full mold growth within 26 days (Figure 10 red), while PVA/PVP/thymol (5 w% thymol) and PVA/PVP/HP-thymol (2.5 w% each) hydrogel coated PE sheets prevent any mold growth. Our findings indicate inferior HP activity against mold. A possible explanation is linked to the fast release rate of HP relative to the moderate release rate of thymol (Figure 4 yellow and orange, respectively). Therefore, most of the HP in the PVA/PVP/HP coating shrivels within the first 2.5 days and runs out after six days, while the thymol shows a moderated and extended release over the 26-day period. This continuous thymol presence in the hay environment ensures the prevention of mold growth.

As both PVA/PVP/thymol and PVA/PVP/HP-thymol coatings completely prevent mold growth, any relative advantage among the two could not be distinguished. Therefore, a stringent complementary test was conducted. To identify the relative mold inhibition concentrations of thymol and HP combined with thymol, coatings were set with lean fungicide content (precursors of 1.25 w% thymol and 0.63 w% each of HP combined with thymol, Figure 11). These lower HP and thymol concentrations challenge their activity against mold development. Hay exposed to PVA/PVP coated PE sheet (control) introduced fully developed mold within 26 days (Figure 11 blue). PVA/PVP/HP-thymol combined coatings enable initial mold growth (20%) within 6 days and full mold development within 26 days (Figure 11 orange). Meanwhile, mold exposed to PVA/PVP/thymol (1.25 w% thymol precursor) enables initial mold development of 2%, followed by growth stagnation over the rest of the test (Figure 11 black). This may be explained by the prolonged exposure to the thymol vapor released from the PVA/PVP/thymol coatings (Figure 5, black). In contrast, PVA/PVP/HP-thymol coatings shortened HP and thymol release durations, resulting in earlier emptying (Figure 5, HP—yellow dotted line, thymol—orange dashed line). This shortage creates conditions that allow the mold to thrive.

These results are directly in line with the HP and thymol release profiles. The following two conclusions are reached by these experiments: (1) PVA/PVP/HP hydrogel coatings are the least effective against mold growth (Figure 10). (2) The combination of entrapped HP and thymol does not contribute synergistic activity against mold development. Furthermore, the coatings loaded with thymol solely were found to be more effective than those also loaded with HP.

## 4. Conclusions and Future Work

PVA/PVP biodegradable hydrogel coatings were loaded with environmentally friendly HP and/or thymol fungicides and coated on PE sheets. The coatings loaded solely with thymol were found to be the most effective for prolonged protection against mold development, even when loaded with a concentration of 1.25 w% thymol (precursor hydrogel solution). These coatings may have environmental and economic implications. The hydrogels biodegrade over time and therefore generate less waste. Moreover, the fungicidal properties are based on thymol, which naturally originates from thymine plants rather than on hazardous synthetic pesticides. HP, the additional pesticide, decomposes into oxygen and water molecules, leaving non-toxic residues. Therefore, both HP and thymol constitute an environmentally friendly supplement.

The coatings with a continuous release of thymol and HP combined with thymol demonstrated complete mold inhibition. Such coatings can save many crops from extermination due to mold spreading. All these advantages make these hydrogel coatings particularly valuable as environmentally friendly and effective hay preservatives. Research is underway in more realistic settings to adapt the coatings to an industrial scale. A full-scale system may present a promising green solution for hay preservation.

These coatings can also be complementary to other mold reduction techniques. One technique is controlling the preharvest growing conditions. For example, bio-monitored earthworms [35,36,37,38] can fertilize crop soils and facilitate the growth of relatively pest-resistant crops [39]. The harvest can be packaged with an anti-mold surface. The higher yield of mold-free crops using an environmentally friendly approach is expected to have a significant economic impact.

## Figures and Tables

**Figure 1 polymers-14-05518-f001:**
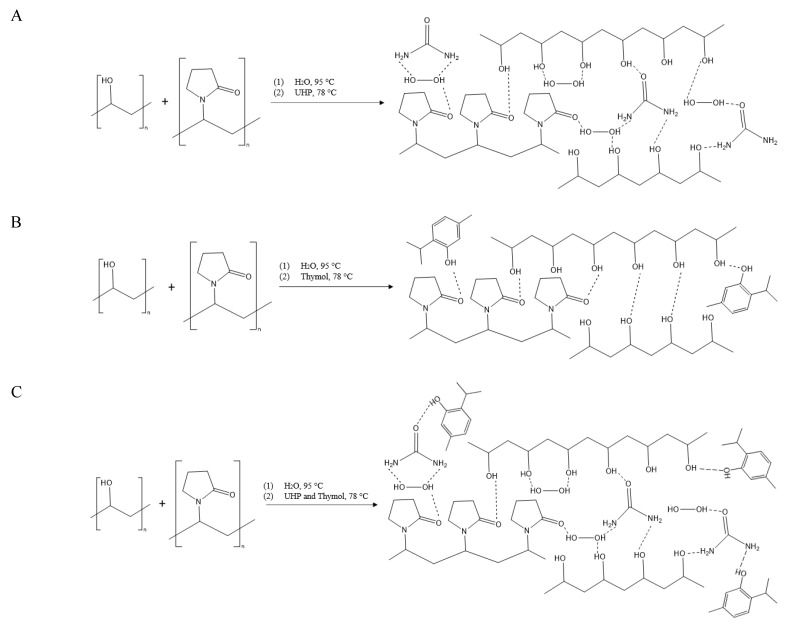
Preparation of PVA/PVP hydrogel with HP (**A**), thymol (**B**) or both (**C**).

**Figure 2 polymers-14-05518-f002:**
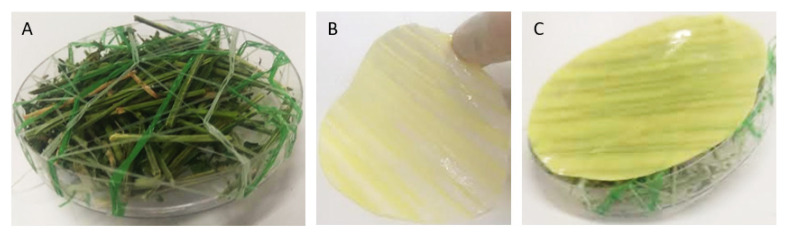
Net covered Petri dish prefilled with water and hay (**A**), hydrogel side of coated PE sheet (**B**) and the covered dish with the hydrogel oriented towards the hay (**C**).

**Figure 3 polymers-14-05518-f003:**
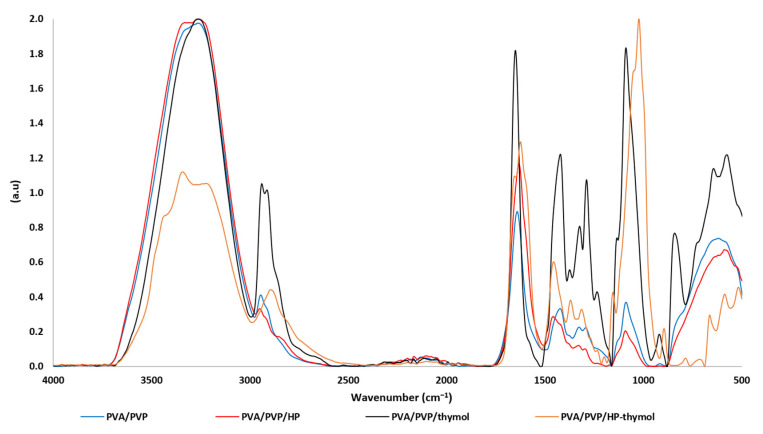
FTIR/ATR spectra of plain (blue), HP-loaded (red), thymol-loaded (black) and HP and thymol-loaded (orange) PVA/PVP hydrogels.

**Figure 4 polymers-14-05518-f004:**
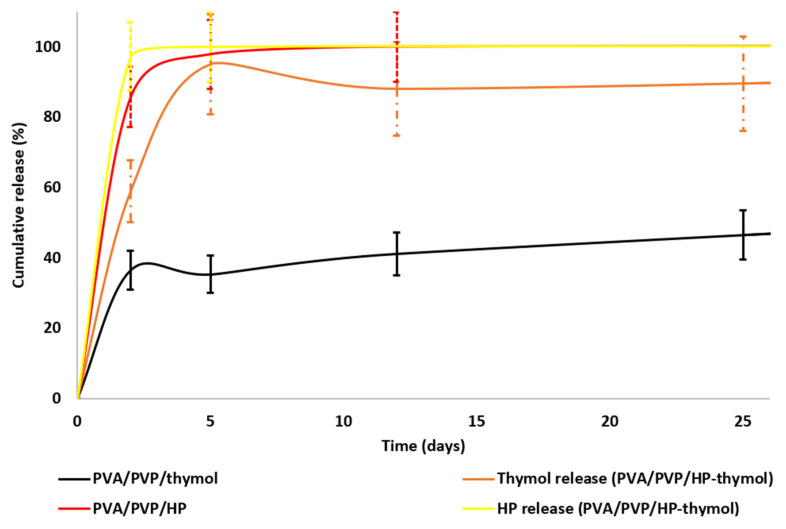
Cumulative release of HP from PVA/PVP/HP (red), thymol from PVA/PVP/thymol (black) and HP (yellow) and thymol (orange) from PVA/PVP/HP-thymol hydrogel coatings to the hay environment. Release rates were measured in triplicate.

**Figure 5 polymers-14-05518-f005:**
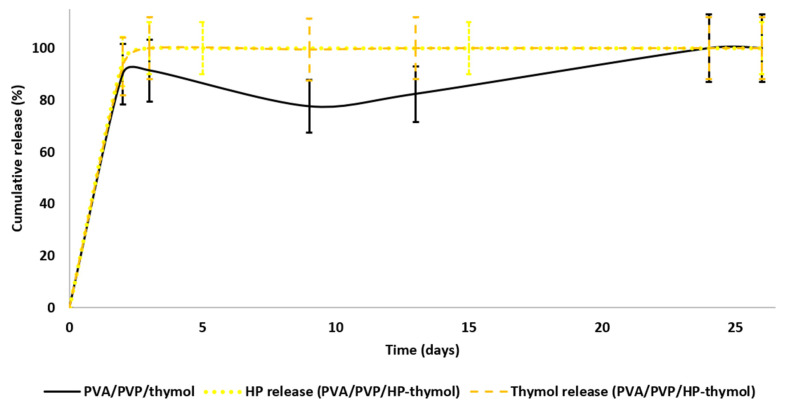
Cumulative release of HP (yellow-dotted line) and thymol (orange-dashed line) from PVA/PVP/HP-thymol (0.63 w% each of HP and thymol) and of thymol (black line) from PVA/PVP/thymol (1.25 w% thymol) to the hay environment (performed in triplicate).

**Figure 6 polymers-14-05518-f006:**
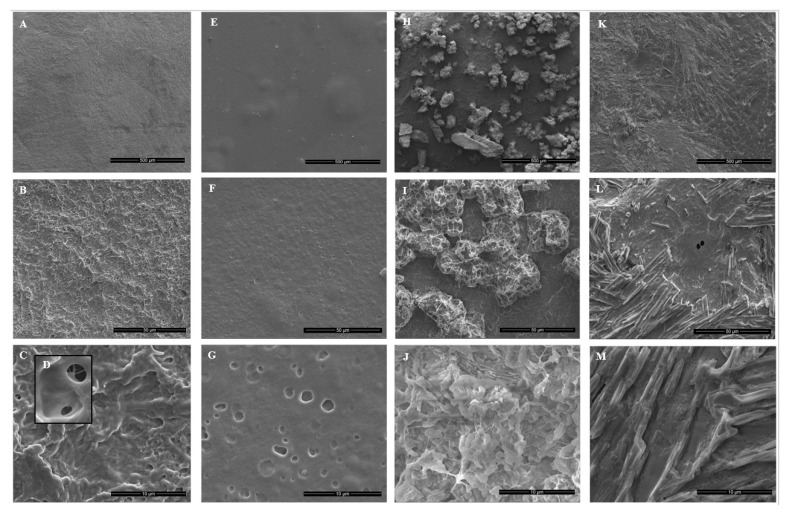
SEM images of neat PVA/PVP × 200 (**A**), 2000 (**B**), 10,000 (**C)** and 40,000 (**D**), PVA/PVP/HP × 200 (**E**), 2000 (**F**) and 10,000 (**G**), PVA/PVP/thymol × 200 (**H**), 2000 (**I**) and 10,000 (**J**), and PVA/PVP/HP-thymol × 200 (**K**), 2000 (**L**) and 10,000 (**M**).

**Figure 7 polymers-14-05518-f007:**
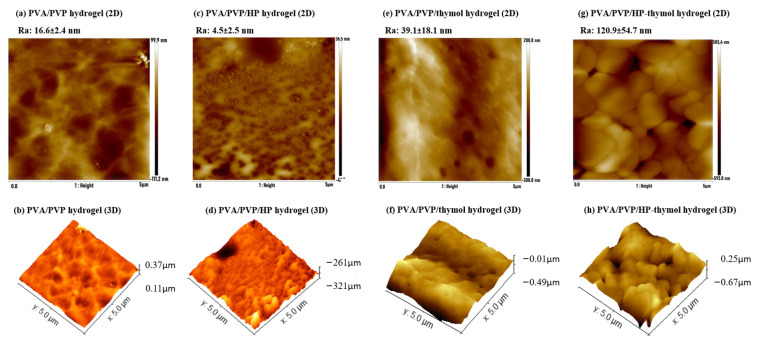
Hydrogel coating average roughness morphology in 2D and 3D views of PVA/PVP (**a**,**b**), PVA/PVP/HP (**c**,**d**), PVA/PVP/thymol (**e**,**f**) and PVA/PVP/HP-thymol (**g**,**h**).

**Figure 8 polymers-14-05518-f008:**
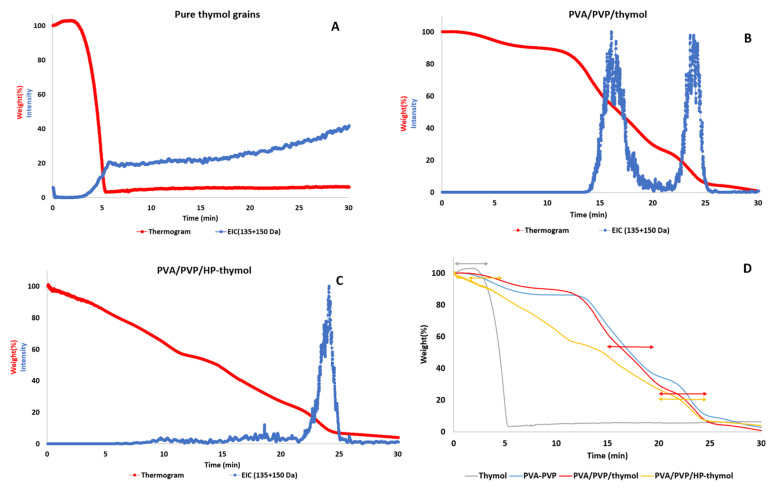
TGA-MS of pure thymol grains (**A**). Comparative graphs of thymol alone and combined with HP entrapped in PVA/PVP hydrogel (**B** and **C**, respectively). Thermograms of weight loss as a function of time (**D**) of thymol (grey), neat PVA/PVP hydrogel (blue) and hydrogels loaded with thymol solely (red) or combined with HP (orange).

**Figure 9 polymers-14-05518-f009:**
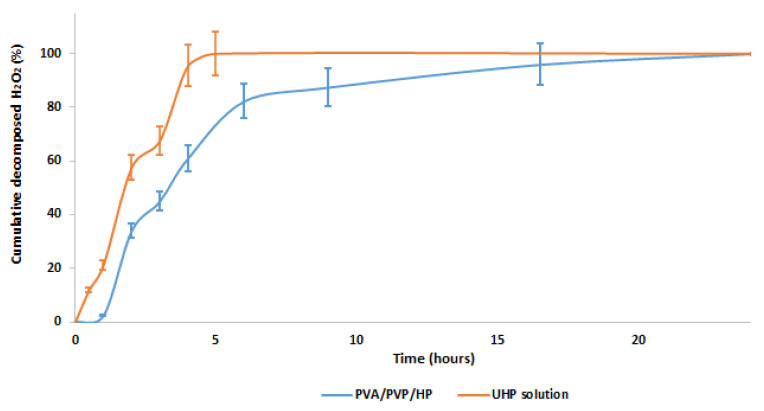
HP decomposition over time in PVA/PVP/HP hydrogel (blue) and aqueous UHP solution (orange).

**Figure 10 polymers-14-05518-f010:**
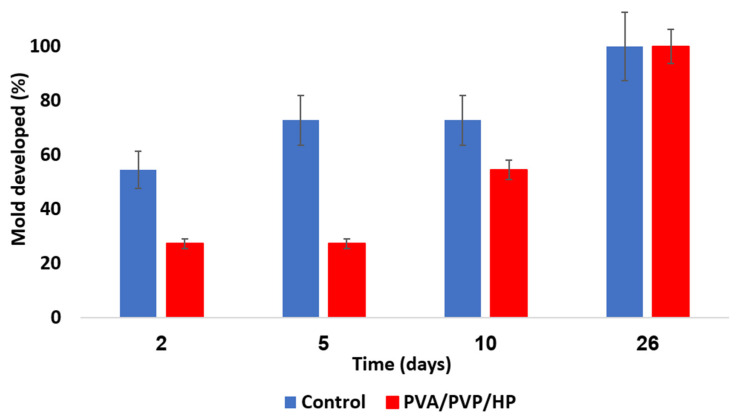
Mold growth rate over several days. Uncoated PE sheets (blue) are a control. PVA/PVP/HP hydrogel coated sheets (red) contain an entrapped anti-mold component.

**Figure 11 polymers-14-05518-f011:**
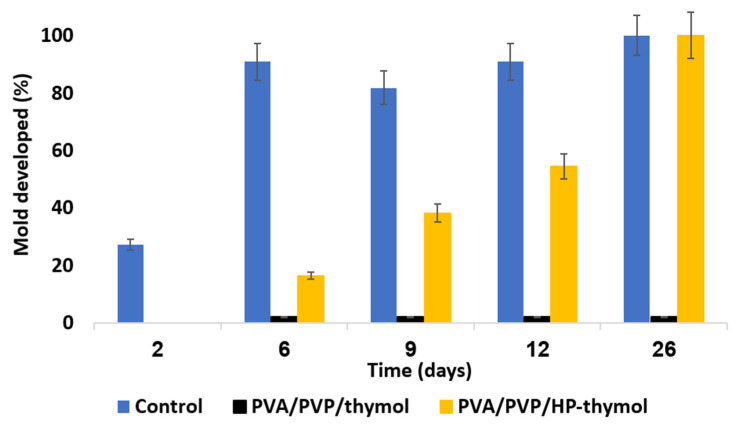
Mold growth rate over several days as a function of coating type. The PVA/PVP hydrogel PE coated sheets (blue) are a control. PE sheets coated with PVA/PVP/thymol (1.25 w% thymol precursor, black) or PVA/PVP/HP-thymol (0.63% each of HP and thymol precursors, orange) present coatings with entrapped anti-mold components.

**Table 1 polymers-14-05518-t001:** HP and thymol initial contents (precursor solutions) in combined hydrogels.

Initial Concentration	2.5 w% Thymol, HP	0.63 w% Thymol, HP
Thymol (w%)	5.9 ± 0.5	0.95 ± 0.04
HP (w%)	5.6 ± 0.5	0.66 ± 0.01

## Data Availability

Not applicable.

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
