# Peer review of "Fabrication and Characterization of Hydrogen Peroxide and Thymol-Loaded PVA/PVP Hydrogel Coatings as a Novel Anti-Mold Surface for Hay Protection"

_polymers, 2022, doi:10.3390/polym14245518_

Round 1

Reviewer 1 Report

The topic of the study fits very well into the topic of the Special Issue “Multifunctional Nanocomposites Used in Agricultural Application, Environmental Treatment” and is very interesting considering its future application. Of course, as the authors mentioned in the Conclusions, it remains to be seen how the coating will be achieved at an industrial level.

The novelty of the method and its advantages are presented.

If there are two or more bibliographic references, they should be placed between the same square brackets.

A separate paragraph should be made on the materials used, before the preparation of the hydrogels (2.1 Materials).

Subsection 2.2.3 (Determination of HP content in PVA/PVP/HP hydrogel coating): it is mentioned that "10 mg coatings were dipped in 10 ml DDW and shaken at room temperature until all the HP was extracted into the water". The exact time to shake should be mentioned, the expression is too general.

More attention is needed when numbering the paragraphs. For example, for simplicity, it is necessary to rearrange and group the Subsections, as follows:

Subsection “2.2.2: Determination of thymol content in hydrogel coatings” should contain the two hydrogel cases: 2.2.2.1 - PVA/PVP/thymol and 2.2.2.2 - PVA/PVP/HP-thymol hydrogel coating, respectively. And the same for HP: Subsection “2.2.3. Determination of HP content in hydrogel coatings” should contain the two cases: 2.2.3.1 - PVA/PVP/HP hydrogel coating and 2.2.3.2 - PVA/PVP/HP-thymol hydrogel coating, respectively.

Subsection “2.2.5: Release rate system” becomes 2.2.4 and must contain the two cases:  2.2.4.1 - Thymol release rate and 2.2.4.2 – HP release rate, respectively, each of them with the two types of hydrogels. Because these two cases (thymol and HP) should be treated together, not separately, as they appear in the current form.

Subsection “3.2. Entrapped HP and/or thymol content”, the expression of concentrations must be revised (% w/w or w% ?). Surely ".178 ±2.4" is wrong, it must be corrected.

In Figure 10, the values for mold growth rate for PVA/PVP/thymol (black) and PVA/PVP/HP-thymol (mustard) are not visible in the graph. If they are zero even after 2 days, they should no longer be mentioned in the figure, only in the text.

Finally, I recommend a revision of the English language, several unsuccessful expressions were noticed, but it is quite difficult for me to mention each one separately, since the lines of the manuscript are not numbered.

Reviewer 2 Report

This work reported an environment-friendly anti-mold fungicide comprising hydrogen peroxide (HP) and thymol entrapped in a polyvinyl alcohol/pyrrolidone (PVA/PVP) hydrogel (PVA is biodegradable and PVP is water soluble non-toxic) coated on a polyethylene (PE) film for preservative hay packaging. The hydrogel improved the thermal stability of the entrapped HP and thymol, resulting in a prolonged release into the hay thereby increasing anti-mold activity. A higher yield of mold-free crops using an environment-friendly approach is expected to have a huge impact on the food economy.  Some interesting results were obtained.  I would recommend the publication of this paper in Polymers after the authors address the following questions.

1.       In Figure 6, different coatings have different morphologies, why?

2.       Whether the thymol can improve the roughness? if so, please prove it in detail.

3.       The combination of entrapped HP and thymol does not contribute synergistic activity against mold development, please explain from the point of view of the mechanism.

4.       Some important references should be cited, Chemical Engineering Journal 429 (2022) 132342; ACS Appl. Mater. Interfaces 2022, 14, 11672−11680; Adv. Optical Mater. 2022, 2200769; Research Volume 2022, Article ID 9838071, 13 pages, https://doi.org/10.34133/2022/9838071

5.       I suggest that authors examine the manuscript carefully, there are many errors in the number of Figures.

Round 2

Reviewer 2 Report

I am satisfied with the revised manuscript offered by the authors.  Therefore, I suggest the publication of this work as it is.